# The Aging Heart: A Molecular and Clinical Challenge

**DOI:** 10.3390/ijms232416033

**Published:** 2022-12-16

**Authors:** Davide Lazzeroni, Andrea Villatore, Gaia Souryal, Gianluca Pili, Giovanni Peretto

**Affiliations:** 1IRCCS Fondazione Don Carlo Gnocchi, 20132 Milan, Italy; 2School of Medicine, Università Vita-Salute San Raffaele, 20132 Milan, Italy; 3Department of Arrhythmology and Cardiac Electrophysiology, Ospedale San Raffaele, 20132 Milan, Italy

**Keywords:** aging, elderly, geriatric cardiology, ischemic cardiomyopathy, heart failure, cardiomyopathies, arrhythmias, molecular mechanisms, oxidative stress, inflammation, microRNAs, telomeres

## Abstract

Aging is associated with an increasing burden of morbidity, especially for cardiovascular diseases (CVDs). General cardiovascular risk factors, ischemic heart diseases, heart failure, arrhythmias, and cardiomyopathies present a significant prevalence in older people, and are characterized by peculiar clinical manifestations that have distinct features compared with the same conditions in a younger population. Remarkably, the aging heart phenotype in both healthy individuals and patients with CVD reflects modifications at the cellular level. An improvement in the knowledge of the physiological and pathological molecular mechanisms underlying cardiac aging could improve clinical management of older patients and offer new therapeutic targets.

## 1. Introduction

Globally, cardiovascular diseases (CVDs) represent the leading cause of mortality and disability among the growing population of older adults [1]. Problems in the field of “geriatric cardiology” are made more complex by the presence of many comorbidities, multiple treatment regimens, frailty, cognitive impairments, and reduced functional capacity, as well as changes in the social environment [2]. Although the burden of most CVDs, such as hypertension, coronary artery disease, and arrhythmias, rises with age, earlier-onset cardiovascular conditions, such as myocarditis and cardiomyopathies, are often investigated later, resulting in more severe conditions in older patients. All those factors should be taken into consideration when making a differential diagnosis [3] (Figure 1). In addition, clinical guidelines often lack specific recommendations for individuals aged ≥ 75 years, as shown in randomized clinical trials [4]. Age-associated alterations in heart structure and function are linked to changes in signaling pathways and gene expression of the cardiac transcriptome, both in senescence and disease [5]. A better knowledge of intracellular modifications and molecular mechanisms could improve therapeutic strategies for cardiac protection [6,7]. This review aims to provide insight into CVDs in the aging population from both clinical and molecular points of view (see Figure 1). 

## 2. Definition of the Aging Heart 

Biological aging is the gradual deterioration of functional characteristics in living organisms. According to the Framingham Heart Study and the Baltimore Longitudinal Study on Aging (BLSA), aging causes an increase in the prevalence of left ventricular (LV) hypertrophy, a decline in diastolic function, and a decline in exercise capacity despite relatively preserved systolic function at rest, as well as an increase in the prevalence of atrial fibrillation in healthy individuals without concomitant cardiovascular diseases [8]. The characteristics of murine cardiac aging closely resemble those of human cardiac aging [8,9] Echocardiography on a mouse lifespan cohort revealed that the left ventricular mass index (LVMI) and left atrial dimension grew considerably with age. In addition, diastolic function, as evaluated by tissue Doppler, decreased with age; however, systolic function only decreased slightly when the older mice were compared with the young adults. The MPI also deteriorated with age, mirroring the age-related reductions in systolic and diastolic performance [9,10]. Furthermore, the relatively short lifetime and the availability of genetically engineered mice are the benefits of using a mouse model in the investigation of the molecular causes of heart aging [9]. Despite possessing comparable cardiac aging characteristics as humans, laboratory mice do not develop increased blood pressure or unfavorable blood glucose and lipid profiles [9,11,12], allowing the intrinsic cardiac alterations of aging to be explored without the extra problems of cardiovascular risk factors such as hypertension and diabetes [9].

## 3. Cardiovascular Risk Factors

One of the most important risk factors for cardiovascular disease (CVD) is age. Indeed, by 2030, almost 20% of the world population will be over the age of 65, which will result in a major rise in CVD prevalence. This emphasizes the significance of comprehending the processes underlying the aging process and its relationship to cardiovascular disease phenotypes (see Figure 2). 


**Age as an independent risk factor**


Age, which is linked to an increased chance of developing a variety of new cardiac risk factors, such as obesity and diabetes, plays a vital role in the deterioration of cardiovascular functionality, resulting in an increased risk of cardiovascular disease (CVD) in older adults [13,14,15]. Furthermore, the prevalence of most types of CVDs is considerably higher among older adults compared with the general population [16]. Throughout an individual’s lifespan, there is an incremental acquisition of several CVD risk factors with age. Nevertheless, age remains an independent risk factor when these risk variables are included in a multivariable regression model [13].


**MicroRNAs**


microRNAs (miRNAs) are involved in the aging process and help to regulate many mechanisms underlying cardiac changes in the elderly [17]. Aging is specifically associated with an increased expression of miR-34a, which is caused by an upregulation of p53 signaling. Indeed, the miR-34 family induces apoptosis, which emphasizes the central role of miR-34a in the mechanisms underlying aging [18]. Moreover, in aged cells, a reduced amount of miR-146a is found. MiR-146a reduces oxidative stress by downregulating the expression of NOX4, which is the major catalytic subunit of NADPH oxidase [19]. Some miRNAs, including the senescence-associated miR-17-92 cluster, have been shown to inhibit apoptosis [20]. Finally, the expression of miR-17, which is reduced by hypoxia, causes a downregulation of Casp9 and apoptotic protease-activating factor 1 (Apaf-1) [21].


**P66^shc^**


It is well known that aging can affect several molecular mechanisms, leading to hypertension and dyslipidemia. Moreover, metabolic disorders, including obesity, diabetes, and insulin resistance, are linked with premature features of vascular and cardiac senescence, pointing out the strong association between aging, metabolism, and cardiovascular disease [22]. This connection might be explained by several factors, such as p66^Shc^. This enzyme leads to the production of reactive oxygen species (ROS) through the oxidation of cytochrome C and, consequently, to the activation of apoptotic mechanisms [23]. 


**mTOR pathway: AMPK**


Growing evidence supports the notion that AMPK plays a crucial role in the regulation of effectors involved in metabolic mechanisms, longevity, and cardiovascular homeostasis [22]. AMPK controls the mTOR pathway through the phosphorylation of the TSC1/2 complex and modulates IGF-1 signaling via the extracellular signal-regulated kinase (Erk) cascade [24]. The fact that the pharmacological triggering of AMPK causes the senescence of vascular smooth muscle cells [25] and the improvement of ROS-driven endothelial dysfunction [26] is particularly important. Moreover, metformin, which is used in diabetes treatment, has been shown to prevent ischemia-reperfusion injury and adverse remodeling of the left ventricle [27]. Considering all the above, AMPK might be considered a therapeutic target to prevent the aging process. 


**NAD-dependent proteins: SIRT1**


Another factor that should be considered is the SIRT1 gene, which is an NAD-dependent protein that protects the heart from senescence, ischemia-reperfusion injury, hypertrophy, and cardiomyocyte apoptosis [28]. In addition, pharmacological activation of SIRT1 by resveratrol causes many benefits, including a decrease in fibrotic collagen deposition, which in turn leads to an improvement of the ejection fraction and fractional shortening [29]. A study showed that SIRT1 improves endothelial function and prevents macrophage foam cell formation and calcification of vascular smooth muscle [30]. SIRT1 can also deacetylate LKB1 and, consequently, activate AMPK, thereby ensuring endothelial integrity thanks to eNOS activity and autophagy [31]. Therefore, impairment of the SIRT1-LKB1-AMPK pathway causes an energy imbalance, cellular stress, and activation of apoptosis mechanisms, which can subsequently lead to vascular aging [32]. 


**NF-κB**


NF-κB represents a crucial intermediary between age-induced myocardial inflammation and fibrosis, and its suppression decreases remodeling and cardiac hypertrophy [33]. Activator protein-1 (AP-1) transcription factor JunD is deeply implicated in age-related disease due to its ability to regulate oxidative stress levels; its importance in the vascular context is supported by the observation that its overexpression can rescue endothelial dysfunction in aged mice [34]. Likewise, JunD expression is reduced in peripheral blood monocytes isolated from aged individuals [22]. It is well known that mTOR takes part in the connection between aging and cardiovascular diseases through the stimulation of oxidative stress and inflammatory responses [35]. Aging is associated with an increase of inflammatory adhesion molecules, including ICA-1 and VCAM-1, which contributes to the initiation and progression of atherosclerosis through enhanced monocyte-endothelial cell interactions [36]. Immunosenescence affects the health and survival of elderly individuals. In particular, senescent T cells can produce a large number of proinflammatory cytokines and cytotoxic mediators, which suggests that they may play a role in cardiovascular disease, including hypertension, atherosclerosis, and myocardial infarction.

## 4. Ischemic Cardiomyopathy

The molecular mechanisms underlying vascular aging are still partially unknown, but the importance of endothelial dysfunction in the context of atherosclerosis and CVD development is clear (see Table 1 and Figure 3). 


**Inflammatory markers and cardiovascular risk**


In the last years, inflammatory markers have emerged as strong independent risk indicators for cardiovascular disease; however, their specificity and predictivity may differ in older people [37]. Interleukin-6 (IL-6) was shown to be a stronger predictor of incident coronary disease [38], stroke, and cardiovascular mortality than C-reactive protein (CRP) [39]. Tumor necrosis factor-α (TNF-α) is another marker of CVD in older patients, but not of stroke [38]. Fibrinogen was not associated with increased CVD risk in people aged ≥ 70 years [40]. One of the key factors in this context is microRNA-217, which accelerates atherosclerosis and coronary lesion development and triggers impaired left ventricular function. On the other hand, microRNA-217 inhibition improves vascular contractile function and reduces atherosclerotic development, which is suggestive of a role as a biomarker of cardiovascular aging in humans [41]. For example, plasminogen activator inhibitor-1 (PAI-1) could promote age-associated thrombosis and atherosclerosis [42]. In addition, dysregulated activation of the renin-angiotensin-aldosterone system (RAAS) accelerates the atherosclerotic process [43]. Age-related impairment of autophagy, which may lead to endothelial dysfunction, arterial stiffness, and vascular pathologies, including atherosclerosis and calcification, is strongly associated with vascular aging [44]. 


**T cells and cardiovascular risk**


It has been suggested that senescent T cells are directly involved in the pathophysiology of atherosclerosis and acute coronary syndrome through the release of several factors, such as IFN-γ, that induce macrophages activation and, consequently, the release of metallo-proteinases that degrade the extracellular matrix [45,46]. These lymphocytes also discharge a great amount of perforin and granzyme, resulting in direct lysis of endothelial and vascular smooth cells [47]. During the process of aging and related ischemic conditions, NAD^+^ levels decrease and lead to nuclear and mitochondrial dysfunctions that result in age-related diseases. It has been demonstrated that restoring NAD^+^ using intermediates, including nicotinamide mononucleotide and nicotinamide riboside, may be a good approach for recovering from ischemic injury and age-associated defects [48]. 


**Telomere shortening and cardiovascular risk**


Progressive telomere shortening and dysfunction are responsible for physiological and pathological aging, including cardiovascular diseases [49,50]. Telomere ablation, as well as length-independent telomere damage, possibly due to oxidative stress, is responsible for age-related cardiac dysfunction [51]. Massive oxidative stress, as seen in cardiac ischemia-reperfusion injury, has been shown to induce telomere damage, with rescue by the clearance of senescent cells [52]. Telomeres also play a role in vascular pathobiology [53]. Circulating leukocytes and atherosclerotic plaque-associated vascular smooth cells have shorter telomeres than age-matched controls [54,55]. Therapeutic strategies for the selective elimination of senescent cells may improve cardiac function in older patients.

**Table 1 ijms-23-16033-t001:** Molecular mechanisms and intracellular modifications underlying ischemic cardiomyopathy (ICM).

Molecular Mechanisms and Intracellular Modifications Underlying Ischemic Cardiomyopathy	Studies
- Increase of cytokines (IL-6; TNFα), microRNA, plasminogen activator inhibitor-1 (PAI-1)	[38,41,42]
- Impairment of autophagy	[44]
- The release of IFN-γ, perforin and granzyme by senescent T cells	[45,46,47]
- Reduction of NAD^+^ levels	[48]
- Telomere shortening and dysfunction	[49,50]

## 5. Heart Failure 

Heart failure (HF) is a clinical syndrome with a prevalence that increases considerably with age. This disease might be considered the result of the interaction between cardiovascular aging and specific risk factors, comorbidities, and disease modifiers [56] (see Table 2 and Figure 4). The aging process relates to various alterations in the vascular system [57] and myocardium, such as elastin fiber degradation and an increase in collagen quantity, that may predispose an individual to HF. Smooth muscle cells also tend to grow and accumulate. These tissue and cellular changes could result in vascular stiffening and an increased afterload for the left ventricle. The aging process also affects the vascular endothelial cells and their capacity to produce NO and other vital peptides. All these alterations contribute to myocardial interstitial fibrosis, calcium deposition, and amyloid accumulations [8]. Similarly, cardiac valves also suffer through the aging process, thereby exacerbating cardiac stress and HF vulnerability [58]. In this context, the role of activin type II receptor (ActRII) ligands, including FSTL3, which is an endogenous inhibitor of ActRII ligands that increases with aging and HF severity in humans [59], might be crucial. It has been proven that systemic ACTRII inhibition improves systolic function in murine age-related HF models [59]. There are several similarities between the pathophysiology of frailty and HF, including many inflammatory markers such as IL-6, CRP, and TNF-α [60]. Remarkably, several microRNAs regulate aging [61], and there is a large overlap between these age-related microRNAs and the microRNAs involved in both HF and inflammation in Toll-like receptor (TLR) signaling [62,63]. Biomarkers involved in extracellular matrix organization, inflammation, and tumor cell regulation were up-regulated in older patients with heart failure with reduced ejection fraction (HFrEF), with a strong association between aging and WAP four-disulfide core domain protein 2 (WFDC2), while pathways associated with tumor proliferation were down-regulated [64]. On the other hand, heart failure with preserved ejection fraction (HfpEF) is very common among older people with other conditions. HfpEF is characterized by chronic, low-grade, systemic inflammation, with the inflammatory milieu differing according to the specific comorbidities present [65]. From a molecular point of view, levels of TNF and its receptors (TNFR1 and TNFR2), interleukin (IL)-6 and IL-8, high-sensitivity C-reactive protein (hs-CRP), pentraxin-3, and the chemokine (C-C motif) ligand 2 (CCL2) are all often raised in individuals with HfpEF. Chronic, low-grade, systemic inflammation may harm cardiac structure and function. Experimental results indicate that increased pro-inflammatory cytokine production increases oxidative stress, drives fibroblast differentiation into collagen-secreting myofibroblasts, and induces extracellular matrix degradation, resulting in increased myocardial stiffness and coronary microvascular dysfunction (CMD). Local inflammation also decreases the availability of nitric oxide (NO) and cyclic guanosine monophosphate (cGMP), leading to hypophosphorylation of the large sarcomeric protein titin, which increases cardiac stiffness and affects diastolic function. Oxidative stress may also be involved in the development of metabolic heart disease, which suggests that inflammation and cardiac dysfunction could be linked in a bidirectional manner [65]. Understanding the different mechanisms underlying HF in the elderly may help to identify potential therapeutic targets.

## 6. Arrhythmias

Aging is associated with an increased prevalence of cardiac arrhythmias, which contribute to higher morbidity and mortality in the elderly [66,67,68,69] (see Table 3 and Figure 5). The incidence of cardiac dysrhythmias, both bradyarrhythmia and tachyarrhythmia, increases with advancing age [70,71,72], with more than 80% of pacemaker implantations in the US needed to relieve symptoms caused by bradycardia and/or chronotropic incompetency from sinus node dysfunction or His-Purkinje disease [70,73,74]. Among tachyarrhythmias, AF is the most common arrhythmia encountered in clinical practice, with a 100-fold higher prevalence in octogenarians (8–10%) compared to those younger than 55 years [68,70,75,76,77,78]. Alterations in the mechanical and electrical cardiac system, as well as energetics and metabolism associated with the aging process, which is exacerbated by comorbidities or use of medications, increase predisposition to cardiac arrhythmias [69,79,80,81]. Deposition of amyloid, lipid, and lipofuscin around the atrial pacemaker tissue contributes to bradyarrhythmia in the aging heart [69,70,74,82,83]. In addition, pacemaker cells within the sinoatrial node and AV conduction fibers are progressively replaced with an extracellular matrix composed of collagen and elastin fibers [84], with up to a 10% reduction of the number of pacemaker cells up in individuals 75 years of age or older compared to young adults [85]. Signaling via *β*-adrenergic receptors also lowers with age, contributing to a diminished heart rate response and heart rate variability and a resultant reduction in aerobic work capacity in the elderly [84,86,87,88,89]. Furthermore, aging-induced degenerative changes to the cardiac skeleton affect areas close to the AV node, His-Purkinje tissue, and bundle branches, thereby delaying conduction and predisposing elderly patients to arrhythmias [90,91]. An increased prevalence of first-degree AV block, mostly secondary to the pathological fibrosis of the conduction system, has been observed in older people. Although largely considered benign, a prolonged PR interval has been associated with increased AF [92,93]. These modifications also increase the prevalence of fascicular and bifascicular block, which are associated with a high risk of subsequent advanced AV block, syncope, and even sudden cardiac death (SCD), especially in the presence of alternating bundle branch block, type 2 or advanced second-degree AV block, or transient third-degree AV block [92,94]. Another study pointed out that since mitochondria are the primary producer of ROS, this organelle could be considered a potential target for free radical damage. Indeed, a general decrease in mitochondrial-encoded gene expression, which is related to mitochondrial genomic DNA deletions [95,96] and mitochondrial loss, followed by reduced mitochondrial function has been observed with age [96,97,98]. Changes in mRNA abundance associated with aging have recently been examined by gene expression arrays [96]. Bodyak et al. [99] found reduced mRNA levels of several transcription factors (e.g., Nkx2.5, GATA-4, JunB) in ventricular cardiomyocytes that might be implicated in aging. However, Lee et al. [100] showed that only 10% of the transcripts in the whole mouse heart demonstrated significant changes in abundance with aging [99], meaning that many age-associated changes in transcript abundance may instead be associated with non-cardiomyocytes, strain differences, or altered transcript abundances associated with isolation procedures. It is therefore critical to consider biological diversity when performing studies of aging [96]. 

It has been demonstrated that electrical and structural remodeling with action potential duration prolongation and connexin remodeling increases the refractoriness of cardiac tissue and slows conduction [74,101,102,103]. Action potential duration and repolarization are delayed in the senescent heart [74,104,105], in part due to the downregulation of K+ currents, including Ca^2+^-activated IK+, transient outward (Ito), and ATP-sensitive K+ channels, and in part due to a delay in Ca^2+^ current inactivation (ICaL) [105,106,107,108]. This delay, along with an increase in sodium-Ca^2+^ exchanger activity, enhances the tendency for Ca^2+^-overload-mediated triggered activity and re-entrant arrhythmias [68,78,109,110,111,112,113,114,115]. A decrease in sarcoplasmic reticulum Ca^2+^-ATPase expression [78,116,117] and post-translational modifications that affect the function of the sarcoplasmic reticulum Ca^2+^-ATPase, phospholamban, and the sarcoplasmic reticulum Ca^2+^-release channel (ryanodine receptor 2) further alter Ca^2+^ homeostasis and the aging heart’s susceptibility to arrhythmias [78,118,119,120,121,122,123]. The effects of age-related changes on cardiac microstructure, including the sarcolemma, cytoskeleton, intercellular gap junctions, cellular geometry, and interstitium, as well as mitochondria [78,124,125], are not well defined and require further studies. 

**Table 3 ijms-23-16033-t003:** Molecular mechanisms and intracellular modifications underlying arrhythmias.

Molecular Mechanisms and Intracellular Modifications Underlying Arrhythmias	Studies
- Accumulation of amyloid, lipid, and lipofuscin, which leads to bradyarrhythmia	[69,70,74,82,83]
- Replacement of pacemaker cells with collagen and elastin fibers	[84]
- Delay of action potential duration and repolarization	[74,104,105]
- Impairment of Ca^2+^ homeostasis	[78,118,119,120,121,122,123]

## 7. Cardiomyopathies

Age-related changes in the heart’s pathophysiology, such as vascular, cellular, and interstitial molecular changes, could result in left ventricular hypertrophy, a general deterioration in organ function, and stress-related cardiovascular illness [126] (see Table 4 and Figure 6). While younger patients tend to be more impacted by DCM and HCM, elderly patients appear to be less afflicted, with only 10% of affected patients being above the age of 65. Restrictive cardiomyopathies are rare in the elderly, while severe and concentric hypertrophy are more commonly associated with hypertrophic cardiomyopathy (HCM) [127]. HCM patients of more advanced age are being increasingly recognized due to greater awareness of this disease and increased use of advanced cardiac imaging in clinical practice [128,129,130]. Exposure to several stressors may cause aggregation of proteins, impair cell viability, and cause pathological conditions, including age-related vascular diseases. To reduce this risk, the cell initiates a mechanism involving molecular chaperones to maintain protein homeostasis. Small heat shock proteins (HSPs) as molecular chaperones prevent aggregation or misfolding of proteins and enable their correct refolding under stress [106,131,132,133]. Among them αB-crystallin (CryaB) [134,135] binds to intermediate filaments and sarcomeric myofibrils, preventing their aggregation during stress [136,137,138]. In terms of function, CryaB phosphorylation has been reported to decrease the ability of this protein to act as a molecular chaperone and to provide protection from oxidative stress [71]. A dysfunction of CryaB could cause various forms of muscular disorder, including restrictive, hypertrophic, and dilated cardiomyopathies, heart failure, and skeletal muscle weakness. The phosphorylation status of CryaB is in dynamic equilibrium under physiological conditions and is usually increased under stress and during aging, although changes in the heart remain unknown [106]. In HCM pathophysiology, disturbances in the physiological protein quality control system (PQS), which is formed by heat shock proteins (HSPs), autophagy/lysosomal, and the ubiquitin-proteasome system (UPS), have been reported. Under conditions of oxidative stress, ROS are found to suppress autophagy, which leads to the accumulation of ubiquitinated proteins and subsequently to cardiac fibrosis and hypertrophy [128]. Furthermore, as reported in recent studies using mice, the knock-out of autophagy-associated genes results in the development of age-related cardiomyopathies. This suggests that continuous constitutive autophagy may play a crucial role in maintaining cardiac structure and function [139]. Furthermore, suppression of the WNT pathway could attenuate age-dependent expression of cardiac dilatation and dysfunction, myocardial fibrosis, and apoptosis in a mouse model of ACM [69,140]. The prognosis of elderly DCM patients has significantly improved over the past 20 years, thanks to advances in pharmacologic treatments and earlier diagnosis [141]. In many patients with HCM, age represents a negative risk marker for sudden death, although they are still more likely to die of non-cardiac competing morbidities [128]. Genetic HCM and DCM are characterized by shorter telomeres in cardiomyocytes [142], with a correlation between hypertrophic phenotype severity and leukocyte telomere length [143]. Myosins and myosin-encoded microRNA networks may explain phenotype differences and could represent putative therapeutic targets in HCM patients [144].

Finally, older patients with a previously healed myocarditis or a subtle chronic active inflammation may suffer from post-inflammatory DCM, which may cause HF and/or ventricular arrhythmias [145].

LMNA-associated cardiomyopathy may be underdiagnosed in older patients with DCM, atrio-ventricular conduction disorder, AF, and ventricular arrhythmias [146].

Amyloidosis, especially wild-type transthyretin (TTR) amyloidosis, is underdiagnosed in older people, for whom “red flags” may be misread [147] and prognosis may be affected by diagnostic delays, despite available treatments [148]. The aggregation of misfolded TTR monomers and deposition of extracellular fibrils is favored by age-related oxidative modifications [149].

In conclusion, the aging of the healthy heart is a complex process characterized by mild cardiomyocyte hypertrophy, increased cellular senescence, and cell replacement within the extracellular matrix; these changes eventually result in a loss of both contractile function and endogenous protection from irreversible injury [106,136]. However, primary genetic and acquired cardiomyopathies should still be considered among differential diagnoses.

## 8. Clinical Management of CVD in Older People

Aging, as well as cardiovascular aging, is a natural and inescapable process; nevertheless, detecting people with accelerated aging is difficult. It is critical to identify patients with accelerated cardiovascular aging and define the mechanisms behind this process to develop preventative interventions targeted at reducing the process of accelerated cardiac and vascular aging. Firstly, it is essential to evaluate and treat each cardiovascular risk factor. This is because the acceleration of cardiovascular system aging primarily depends on the harmful role of both traditional and emerging cardiovascular risk factors, such as familial history, arterial hypertension, dyslipidemia, diabetes, obesity, smoking, and unhealthy lifestyle [150]. Secondly, it is necessary to perform cardiological evaluations to identify cardiac organ damage (hypertrophy, dilation, systolic and/or diastolic dysfunction) and vascular organ damage (atherosclerosis or vascular stiffness) using a multiparametric diagnostic approach [151] (Figure 7). Similarly, non-drug therapies, such as changes in diet and activity, represent the cornerstone of anti-aging medicine. Moreover, identifying the optimal medical therapy for specific cardiac diseases represents the first step toward slowing down accelerated cardiovascular aging [152]. Finally, novel diagnostic (genome, miRNome, transcriptome and metabolomics) and therapeutic tools, such as cytokine antagonists, TGF-β inhibitors or endothelin and VEGF inhibitors [153], are also emerging as potential methods to slow down accelerated cardiovascular aging (Figure 7). 

Older adults, especially those aged ≥ 75 years and with multiple disabilities, are underrepresented in most cardiovascular clinical trials, resulting in knowledge gaps related to cardiovascular care for this population [4]. In addition, there is great heterogeneity and biological diversity in this population, which are independent of age [152]. Adopting a patient-centered-approach, which considers individual comorbidities, life expectancy, cognitive function, frailty, and patient preferences, is critical for establishing the optimal management strategy [152]. According to recent guidelines, reducing blood pressure to a cut-off of <140/90 mmHg is recommended for older adults suffering from hypertension, and a further lowering to 130 mmHg should be considered in individuals aged ≥ 70 years [150]. Aspirin in primary prevention did not demonstrate a reduction in CVD, and increased major bleeding risk in individuals aged ≥ 70 years [154]. A statin-based therapy for primary prevention in older adults aged ≥ 70 years who have a high 10-year CVD risk, as estimated by the SCORE2-OP algorithm, has been proposed [150,155]. Recently, a polypill containing aspirin, ramipril, and atorvastatin was proven to be effective in secondary prevention in adults aged ≥ 75 years [156]. Aging is a risk factor for both ischemic and bleeding events, but the need for antithrombotic therapies is increased in older people, mainly due to atrial fibrillation (AF). New oral anticoagulants (NOACs) have shown better efficacy and safety than warfarin with regards to reducing stroke, all-cause mortality, and intracranial hemorrhage, even in patients aged ≥ 75 years [157]. However, due to competing mortality risks, other studies failed to confirm the net clinical benefit of anticoagulation for AF in older patients [158]. Invasive procedures, such as revascularization and transcatheter valve interventions, have been proven to reduce major cardiovascular adverse events (MACE) and mortality, without additional major bleeding risk, in patients aged ≥ 75 years [159,160,161,162,163,164]. Palliative care and treatment discontinuation, based on the evaluation of life quality, symptom burden, and disease acceptance, are often neglected in CVD [165]. However, patients with end-stage heart failure (HF) could benefit from this kind of intervention [166].

## 9. Molecular Therapies


**mTOR pathway inhibitors**


A recent study by Infante et al. [167] showed that the use of the mTOR inhibitor everolimus in kidney transplant recipients dramatically reduced CVD risk by reducing levels of inflammaging markers, namely serum pentraxin-3 and p21ink, and improving mitochondrial function/biogenesis in PBMC, resulting in more efficient oxidative phosphorylation, antioxidant capacity, and glutathione peroxidase activity [167]. Further supporting these antioxidant and anti-inflammatory effects of rapamycin, pathway analysis revealed an upregulation of free radical scavenging genes and a downregulation of NF-κB signaling genes after rapamycin treatment in adult stem cells [168]. A possible effect of berberine, a Chinese herbal medicine, on both the mTOR pathway and AMPK has been reported [169]. A novel possible therapy to modulate the mTOR, AMPK, and sirtuin pathways comes from the CALERIE trial, which suggests that a short-term calorie restriction (CR) of 25% is effective at delaying age-related phenotypes and improving CVD risk factors in adults [170,171,172]. 

While age-related increases in superoxide production are associated with increased expression and activity of NADPH oxidase [96], CR appears to constrain this source of ROS, with the expression of NOX4 and the p67 subunit of NOX2, as well as the activity of NADPH oxidase, being reduced in old mice after CR [172]. Pharmacologically, AMPK activity can be increased directly after treatment with aminoimidazole carboxamide ribonucleotide (AICAR, an adenosine analog) and indirectly after metformin treatment. It has been shown that long-term metformin treatment can increase an individual’s life span [173]. Direct AMPK activation by AICAR has also been shown to increase tissue antioxidant defenses, including increasing skeletal muscle expression of MnSOD [174]. Moreover, several studies have shown that activating AMPK can lower inflammatory cytokines, and that this is linked to muted NF-κB signaling in a range of tissues, including endothelial cells [172,175].


**SIRT1 stimulation**


Genetic models provide direct evidence for a protective role of SIRT1. In particular, it has been shown that cardiac-specific SIRT1 overexpression leads to cardiac protection against ROS and delays age-related cardiac phenotypes [172,175]. One such small molecule activator of SIRT1, SRT1720, has recently been shown to increase life span [172,174] and improve metabolic function in aged mice [176]. Furthermore, treatment with SRT1720 seemed to reverse age-associated NF-κB activation and reduced arterial cytokine expression in old mice [177], consistent with the effects of CR on arterial inflammation [172]. Resveratrol, a plant polyphenol, exhibits antiaging, antitumor, and vascular protection effects by enhancing the binding of SIRT1 and LKB1 and subsequent SIRT1 activation. In this way, resveratrol, through LKB1-dependent SIRT1 activation could increase mitochondrial biogenesis and respiration [178].


**Telomere-related therapies**


Telomeres are repetitive DNA sequences located at the extremities of chromosomes [179]. Telomeres get shorter as we age in most of our tissues, contributing to the organ and tissue failure we see as we age [179]. Telomerase is a reverse transcriptase that adds new telomeric repeats to short telomeres and prevents them from triggering apoptosis or cellular senescence [179,180,181]. Healthy lifespan is also positively correlated with longer telomeres in humans, as not smoking and not being obese at the age of 71 were shown to be the most significant factors associated with survival in men aged 85 years or older [180,182]. Patients suffering from age-related diseases and premature aging syndromes display shorter telomeres compared to healthy individuals [183]. A vast number of studies have shown how genetically engineered mice with an overexpression of telomerase had dramatically increased lifespans [179]. While telomerase gene transfer therapy provides an attractive method for cardiovascular restoration and deserves future investigations, many studies seem to agree that a combination of exercise, healthy diet, low everyday stress, and anti-inflammatory agent intake may be beneficial in promoting human longevity by modulating the telomere system and slowing down the effects of many chronic disorders [180,181]. 


**MicroRNAs inhibition**


MiR-217 is a biomarker of vascular aging and cardiovascular risk, as it regulates an endothelial signaling hub and downregulates a network of eNOS, including VEGF, which results in diminished eNOS expression [184]. A recent study by De Yebénes et al. [184] found out that the inhibition of endogenous vascular miR-217 in apoE^−/−^ mice improved vascular contractility and diminished atherosclerosis, highlighting the therapeutic potential of miR-217 inhibitors.

## 10. Conclusions

Cardiovascular disorders of the aging heart are a difficult pre-clinical and clinical problem. The complexity of cardiac problems in elderly people is not explained by metabolic remodeling, loss of proteostasis, DNA instability and telomere shortening alone, but also by epigenetic transcriptome modifications by microRNAs. Pre-clinical and clinical research demonstrates that dietary restriction with adequate intake of specific nutrients, as well as regular exercise, stress management, and smoking cessation, are effective ways to prevent or delay the accumulation of molecular damage that results in tissue degeneration and cardiometabolic dysfunction. With the growing impact of aging, it is essential to reassess CV research, including the increased use of real-world studies to measure long-term effects. Clinical decision-making should integrate molecular and genetic indicators, pointing to personalized therapy. Remarkably, the identification of new molecular targets, as well as improved clinical characterization of older patients, may enhance knowledge and therapy of the aging heart. Furthermore, the use of pharmacological treatments and other interventions should be based on both the patient’s quality of life and preferences, with the appropriate strategy being defined by a multidisciplinary team that includes the individual patient in the decision-making process.

## Figures and Tables

**Figure 1 ijms-23-16033-f001:**
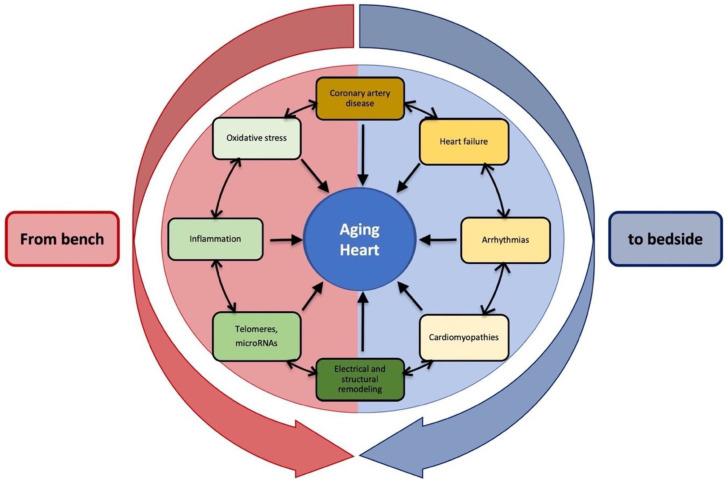
Aging heart: clinical and molecular features of cardiac conditions in older patients. Any of these items could contribute to cardiac aging individually or in combination.

**Figure 2 ijms-23-16033-f002:**
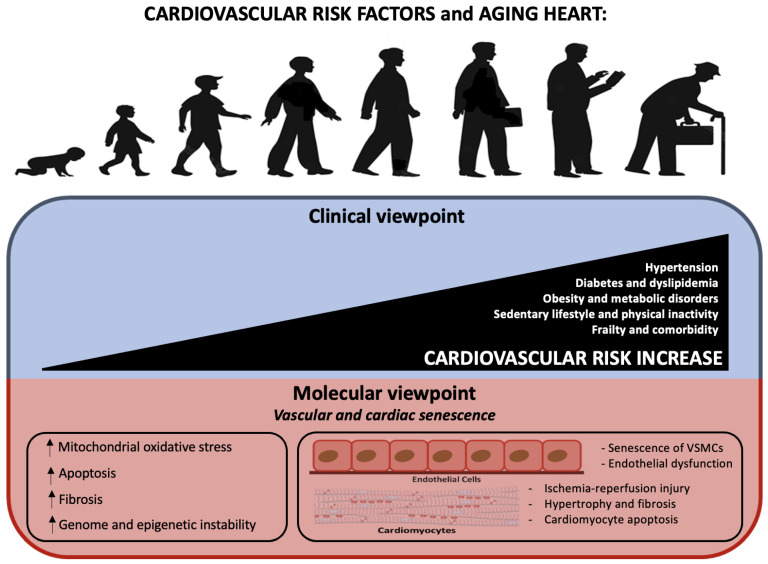
Aging heart and cardiovascular risk: from clinical to molecular viewpoint. The upper figure shows the correlation between age and CV risk, which is explained by an increase in the number of pathologies such as diabetes and dyslipidemia. Underneath, the same events are explained from a molecular point of view.

**Figure 3 ijms-23-16033-f003:**
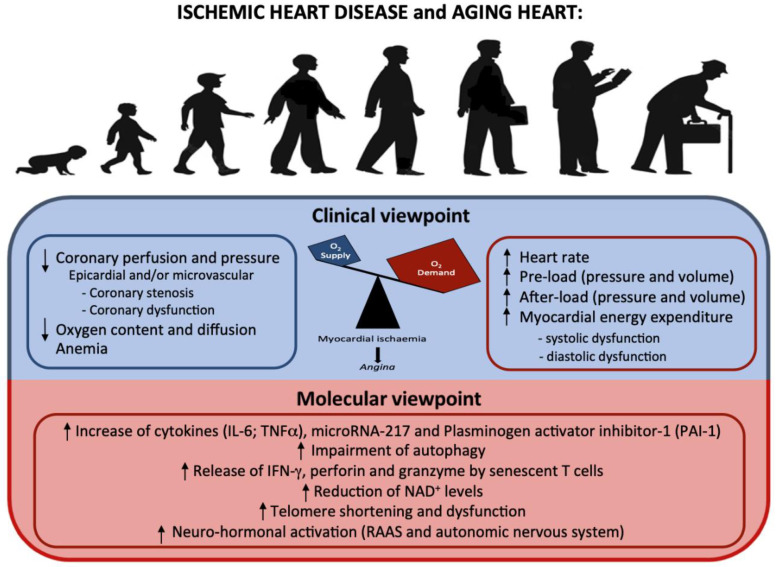
Aging heart and ischemic heart disease: from clinical to molecular viewpoint. An increase in O_2_ demand in the context of a diminished O_2_ supply leads to myocardial ischemia. The pathophysiological factors that cause these alterations are listed in the upper figure. In the lower figure, the molecular modifications secondary to the increased cellular stress are listed.

**Figure 4 ijms-23-16033-f004:**
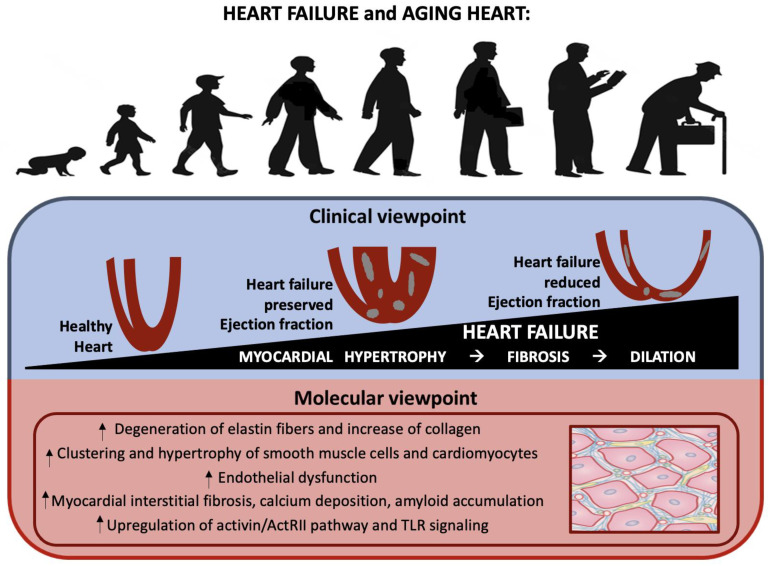
Aging heart and heart failure: from clinical to molecular viewpoint. Age-related myocardial hypertrophy due to multiple mechanisms (lower figure) results in fibrosis and subsequent cardiac dilation (upper figure).

**Figure 5 ijms-23-16033-f005:**
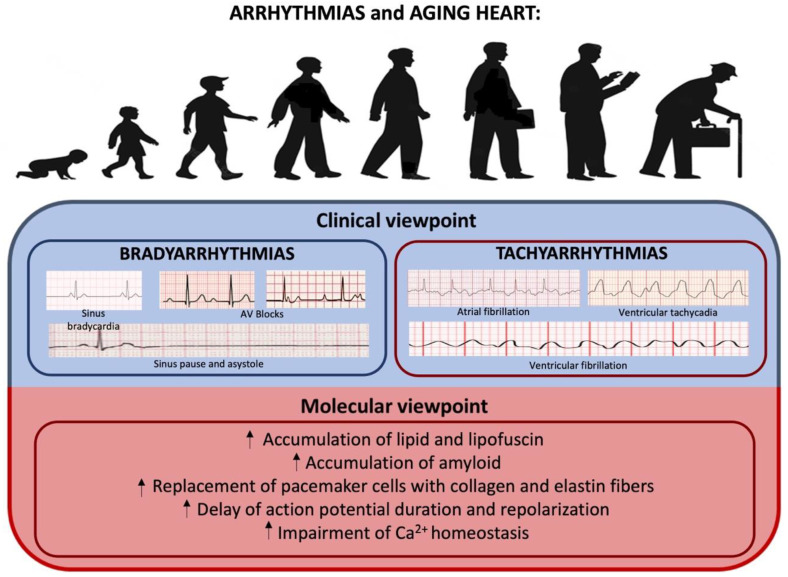
Aging heart and arrhythmias: from clinical to molecular viewpoint. The aging process may be one of the main promoting factors for bradyarrhythmias and tachyarrhythmias.

**Figure 6 ijms-23-16033-f006:**
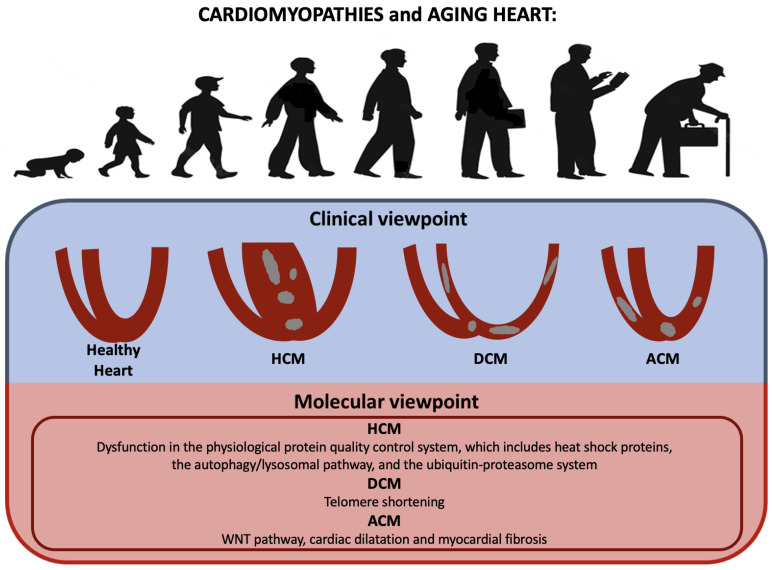
Aging heart and cardiomyopathies: from clinical to molecular viewpoint. Primary cardiomyopathies may be diagnosed even in older people and show specific age-related features.

**Figure 7 ijms-23-16033-f007:**
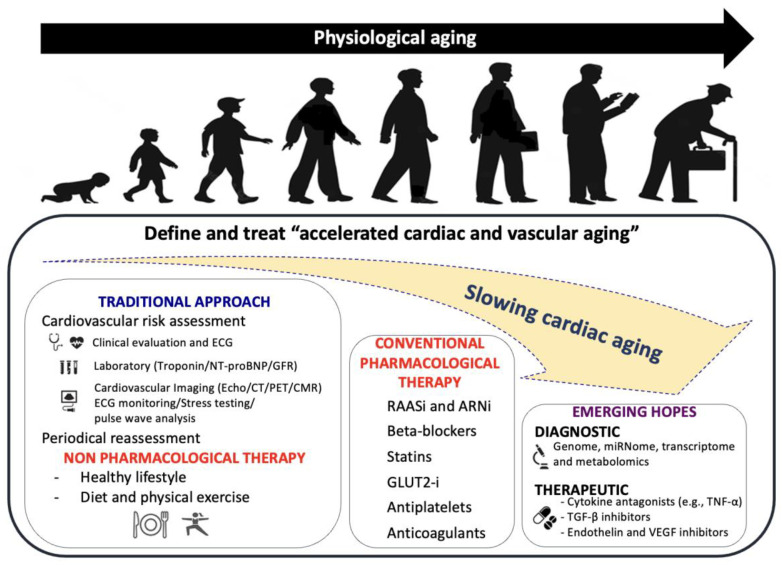
Clinical management of CVD in older patients. From traditional approaches, such as cardiovascular risk assessment and conventional pharmacological therapies, we are moving towards a new era of diagnostic and therapeutic opportunities, as shown in the figure. Detection and treatment of accelerated aging patients might significantly improve outcomes and delay the aging process.

**Table 2 ijms-23-16033-t002:** Molecular mechanisms and intracellular modifications underlying heart failure (HF).

Molecular Mechanisms and Intracellular Modifications Underlying Heart Failure	Studies
- Degeneration of elastin fibers and increase in collagen	[57]
- Clustering and hypertrophy of smooth muscle cells	[57]
- Endothelial dysfunction, which affects the production of NO and other peptides	[57]
- Myocardial interstitial fibrosis, calcium deposition, and amyloid accumulations	[8]
- Upregulation of the activin/ActRII pathway and TLR signaling	[59,62,63]

**Table 4 ijms-23-16033-t004:** Molecular mechanisms and intracellular modifications underlying cardiomyopathies.

Molecular Mechanisms and Intracellular Modifications Underlying Cardiomyopathies	Studies
- Dysfunction in the physiological protein quality control system, which includes heat shock proteins, the autophagy/lysosomal pathway, and the ubiquitin-proteasome system (HCM)	[128]
- WNT pathway, due to its correlation with cardiac dilatation and myocardial fibrosis (ACM)	[69,140]
- Telomere shortening (HCM and DCM)	[142]

## Data Availability

Not applicable.

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
