# Peer review of "The Aging Heart: A Molecular and Clinical Challenge"

_ijms, 2022, doi:10.3390/ijms232416033_

Round 1

Reviewer 1 Report

The article by Lazzeroni and co-workers reviews the literature on epidemiology, etiology, physiology, pathophysiology, and therapeutic prevention of cardiovascular aging. Given the worldwide increase in the aging population and associated increase in older patients with cardiovascular diseases, this review is timely and should provide not only guidelines for age-specific treatments but also for new pharmacological developments targeting signaling pathways putatively involved in cardiovascular aging. I have some major comments which should be addressed by the authors:

11)      The author should discuss existing scientific evidence demonstrating that age constitutes an independent risk factor for cardiovascular disease.

22)      In a brief paragraph, the authors may want to summarize the concept of the ‘aging heart’. This summary should provide, besides a general definition of biological senescence, information on age-specific structural and functional alterations of the cardiovascular system, on existing animal models to study cardiovascular aging and their implications for our current understanding of the cardiac aging process.

33)      Some sections provide information that does not match a section’s heading. For example, lines 186-100 in the section 5 (Arrhythmias) mention changes in transcription occurring in the aging heart but their relationship to arrhythmia specifically is not discussed. There are multiple other examples for a mismatch between section content and section heading which should be corrected if possible.

44)      On numerous occasions throughout the manuscript clear distinctions between what has been scientifically proven and what is being hypothesized by the authors are missing. For example, in lines 136 to 141, changes in the extracellular matrix proteins elastin and collagen as well as in smooth muscle cell properties are suggested to underlie ‘vascular stiffening’ and ‘increased afterload for the left ventricle’. No reference is listed. Similarly, reduced endothelial NO production is implicated in heart failure pathogenies in the elderly, but again a reference is missing. The authors should explicitly distinguish between established evidence and hypothetical mechanisms for cardiovascular aging throughout the manuscript.

55)      The literature is not up to date. For example, refs. 143 and 145l list clinical studies reports published 21 and 12 years ago, but there are clearly more recent studies on these topics. The year of publication is missing from some references.

Reviewer 2 Report

This is a review article of the aging heart.  It is an interesting challenge to connect biological research and clinical manifestation of the heart.  However, there is still a big gap between molecular level research and clinical practice.

The authors should correct punctuation in the abstract.

All the figures require legends to explain.  

Figure 1 must be fully explained or corrected, as the readers may think each item (such as heart failure, arrhythmias, and cardiomyopathies) are related in the order of this cycle. Arrhythmia, CAD, and the other cardiomyopathies could become cause of heart failure individually or in combination.

Each subsection (such as 2. Cardiovascular risk factors, 3. Ischemic cardiomyopathy, etc.) should be written in separate paragraph regarding each molecule.  As each subsection is not organized, it is hard to understand.  The authors may want to use sub-subtitle to organize the paragraph.

Figure 2 also needs legend. What does the white arrow mean? Hypertension does not always come after diabetes and dyslipidemia.

In the section "4. Heart failure", the authors focus on HFrEF only.  They should refer to the other types of heart failure.

In Figure 5, VF and VT are not always caused by aging. 

In the section "6. Cardiomyopathies", HCM and DCM are not always related to aging.  These are observed in younger patients or middle-aged patients.

In Figure 7, is "Slowing accelera- ng aging" typo? Figure legend is required.

"Conclusions" should be revised so that the readers can understand.

Reviewer 3 Report

In the present review the authors discuss the molecular mechanisms behind the cardiac ageing and the cardiovascular diseases (CVD), the clinical management of CVD in aged people and the molecular therapies aimed to delay age-related phenotypes and CVD risk.

However, the topics are dealt with superficiality and lack of depth. The paper is written in extremely poor English. Grammatical and syntactical errors exist throughout the manuscript, which requires significant modification and editing for clarity. Many of these errors distort what the authors would like to say and create many difficulties in the reading comprehension. Moreover, the molecular mechanisms behind the cardiac ageing and the cardiovascular disease are dealt with superficially and may require further improvement. Several figures are not of sufficient quality for review. In some instances, references are not present in the test (e.g. 43 and 44). In light of above deficiencies, the manuscript is not suitable for publication in MDPI IJMS.

Round 2

Reviewer 1 Report

Overall, the authors have responded satisfactorily to my previous comments. A list of abbreviations should be added the manuscript. 

Reviewer 2 Report

The v2. manuscript is easier to read than v1. but is still has grammatical errors throughout the manuscript. The writing style is very complicated and difficult to comprehend.

This article seems to be a review of aging heart.  Accumulation of molecular level research and hypotheses may be interesting to some readers but not really practical for physicians.  Most of the figures may be interesting for non-medical professionals, but they are not quite correct and therefore misleading.

Conclusion is still confusing.  Overall, this article does not reach the high level of publication in IJMS.
